# DIPO: Dual-State Images Controlled Articulated Object Generation Powered by Diverse Data

Ruiqi Wu[1,2,3*]  Xinjie Wang[3]  Liu Liu[3]  Chunle Guo[1,2]  Jiaxiong Qiu[3]  Chongyi Li[1,2†]
Lichao Huang[3]  Zhizhong Su[3]  Ming-Ming Cheng[1,2]
[1]NKIARI, Shenzhen Futian  [2]VCIP, CS, Nankai University  [3]Horizon Robotics

## Abstract

We present **DIPO**, a novel framework for the controllable generation of articulated 3D objects from a pair of images: one depicting the object in a resting state and the other in an articulated state. Compared to the single-image approach, our dual-image input imposes only a modest overhead for data collection, but at the same time provides important motion information, which is a reliable guide for predicting kinematic relationships between parts. Specifically, we propose a dual-image diffusion model that captures relationships between the image pair to generate part layouts and joint parameters. In addition, we introduce a Chain-of-Thought (CoT) based **graph reasoner** that explicitly infers part connectivity relationships. To further improve robustness and generalization on complex articulated objects, we develop a fully automated dataset expansion pipeline, name **LEGO-Art**, that enriches the diversity and complexity of PartNet-Mobility dataset. We propose **PM-X**, a large-scale dataset of complex articulated 3D objects, accompanied by rendered images, URDF annotations, and textual descriptions. Extensive experiments demonstrate that DIPO significantly outperforms existing baselines in both the resting state and the articulated state, while the proposed PM-X dataset further enhances generalization to diverse and structurally complex articulated objects. Our code and dataset are available at https://github.com/RQ-Wu/DIPO.

## 1   Introduction

Articulated objects are pervasive in everyday environments. Achieving accurate modeling of articulated structures is the key enabler for building interactive virtual environments. It plays a crucial in simulation [51, 44, 48], animation [50, 5, 33, 21], robot manipulation [11, 9, 28, 32], and embodied AI [19, 36, 31, 20, 16].

However, constructing such models manually is highly labor-intensive and unscalable. As a result, increasing attention has been devoted to developing automatic methods for articulated object modeling [41, 22, 45, 18, 24, 6, 23]. Despite promising progress, existing methods exhibit clear performance degradation when applied to structurally complex or visually ambiguous objects. These limitations stem from two fundamental bottlenecks.

The first issue is **input modality constraints.** Reconstruction-based approaches [41, 22, 45] often rely on multi-view or multi-state images to reconstruct articulation behavior with high accuracy. While effective, these methods demand expensive data acquisition setups, precise camera calibration, and well-aligned temporal input, making them difficult to scale. On the other research line, benefiting from the controllability of diffusion models [12, 37, 35, 30, 47, 53, 52, 38, 39], many generation-based methods [18, 24, 6, 23] are proposed. They utilize minimal input, such as category priors

---

*This work was done while Ruiqi Wu was a Research Intern with Horizon Robotics.
†denotes correspondence author.

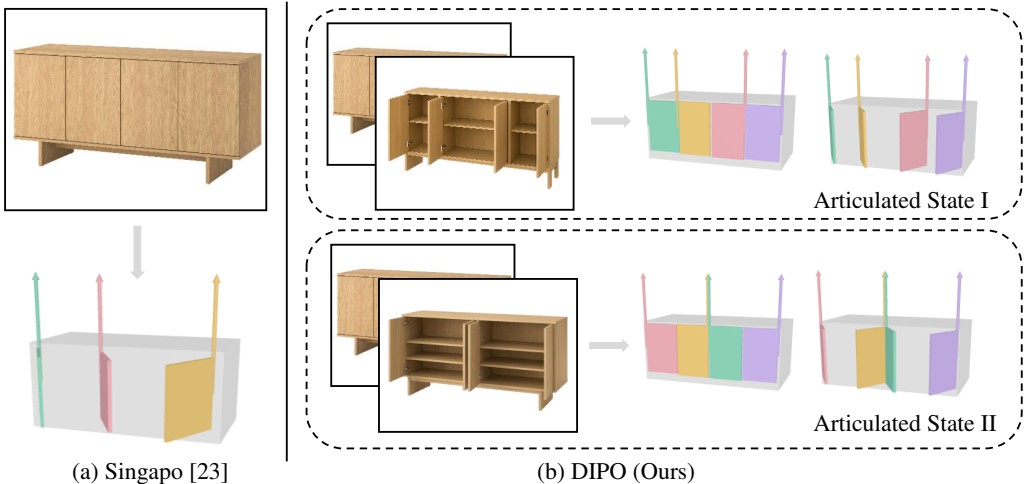

|     | (a) Singapo [23] |     | (b) DIPO (Ours) |     |

Figure 1: **Visual comparison on real-captured data.** (a) SINGAPO struggles with challenging data and fails to model motion relationships due to its reliance on a single input. However, our DIPO (b), which conditioned on dual-state image pairs, effectively generates accurate layouts and enables precise control if part motion across different articulated states.

or a single RGB image, to synthesize articulated objects directly. However, category priors lack spatial specificity, and single-image inputs lack of explicit articulation information. As a result, these methods can only infer kinematic behaviors in a probabilistic manner. Consequently, neither class of methods offers both control and generalization ability when facing challenging data.

Secondly, **limitations in training data.** Data-driven modeling approaches require large-scale datasets with both articulation diversity and structural complexity. However, most existing datasets fall short in some aspects. For example, PartNet-Mobility (PM) [48] offers a large number of articulated assets, but the object instances are dominated by simple and repetitive layouts with limited variability. In contrast, the Articulated Container Dataset (ACD) [14] contains more realistic and structurally diverse objects, but suffers from small scale, limiting its utility for model training.

To address the first issue, we propose **DIPO**, a generation framework for 3D articulated objects conditioned on resting (closed) state and articulated (open) state image pairs. The dual-state image pair encodes essential motion cues and connectivity information. Compared to single-image methods, dual-state input resolves ambiguity in part motion and spatial relationships. As for multi-view methods, it is significantly easier to acquire while maintaining sufficient articulation information. DIPO is built on a diffusion transformer architecture [30] and consists of two core components. First, a *Dual-State Injection Module* helps the network to model the relationships between dual-state images. Second, a *Graph Reasoner* based on Chain-of-Thought (CoT) techniques [43, 17] infers part connectivity step by step. Moreover, this module few-shot learns on visual prompts synthesized by GPT-4o [3, 1] to acquire better performance. The proposed method achieves higher controllability and improved performance in articulated 3D object generation.

In response to the second challenge, we propose a new dataset named **PartNet-Mobility-Complex (PM-X)**, which provides diverse and structurally complex articulated objects with rendered images, URDF annotations [34], and language descriptions. PM-X is built by a fully automated data construction pipeline based on an agent system, named **LEGO-Art**. Starting from natural language prompts sampled from a LLM [3], the pipeline first generates coarse part layouts in a discretized 3D space. Then we develop a toolkit to transfer them to annotations with precise coordinates and articulation parameters. Based on retrieval algorithms [24], we can acquire the final 3D object and the rendered images. Finally, a vision-language model (VLM) [1] is used to filter implausible samples.

We collect a resting state image from the Internet and generate corresponding articulated state images by a visual generative model [1]. As illustrated in Figure 1, our method outperforms the state-of-the-art method, i.e. SINGAPO [23]. Our main contributions are summarized as follows:

- We propose a novel dual-state image model for controllable articulated 3D object generation, integrating layout diffusion and CoT-based connectivity reasoning.

- We develop LEGO-Art pipeline to construct structurally diverse articulated objects, and contribute PM-X, a new large-scale dataset with rendered images and physical annotations.
- Extensive experiments demonstrate that DIPO significantly outperforms state-of-the-art methods, and the proposed LEGO-Art and constructed PM-X dataset enhance generalization to complex structures.

## 2 Related Work

### 2.1 Articulated Object Creation

Recent progress in articulated object modeling can be broadly categorized into reconstruction-based and generation-based approaches.

**Reconstruction methods** commonly rely on multi-view or multi-state inputs to reconstruct part-level geometry and articulation parameters. CLA-NeRF [41] reconstructs articulated objects from sparse multi-view RGB images within a known category. PARIS [22] extends this setting to unknown categories with dual-state multi-view RGB images. Weng et al. [45] further incorporate depth information to support richer geometry priors. However, they rely on densely aligned inputs and known part counts, limiting their applicability in real-world settings. In contrast, our approach only conditioned a pair of images, which reduces input complexity while preserving articulation fidelity.

**Generative approaches** aim to synthesize articulated objects from compact inputs, bypassing the need for dense observations. NAP [18] parses layouts and articulation parameters into graphs and generates articulated 3D objects unconditionally. CAGE [24] achieves a controllable generation from the given articulation graph. Despite these models support efficient sampling, they lack explicit visual guidance to achieve more accurate controllability. URDFormer [6] solves this issue by combining a visual detector [25, 46] to extract spatial layout and a transformer to predict articulation parameters. SINGAPO [23] proposes a diffusion model [12, 37, 35, 30] conditioned on resting state images to generate articulated objects. However, the controllability of current approaches remains limited due to the absence of explicit articulation dynamics. The proposed DIPO effectively addresses this limitation by utilizing the motion information provided by a pair of images captured in the resting and articulated states.

### 2.2 Synthetic Articulated Object Datasets

The availability of large-scale 3D datasets with part-level structures has significantly facilitated research on articulated object modeling. Early datasets such as those used in [13, 49] are constructed by manually segmenting shapes from ShapeNet [4] and SketchUp [40], and annotating articulation parameters for part pairs. Shape2Motion [42] expands the scale by introducing an annotation tool that supports visual verification through animation. PartNet-Mobility[48] is a large-scale articulated object dataset constructed on PartNet[27]. It offers annotations of part-level articulation along with high-quality rendered images, and is one of the most widely adopted benchmarks. GAPartNet [10] focuses on functional part detection across categories, emphasizing generalizable and actionable parts such as buttons and handles. These datasets have enabled the development of deep learning models for articulation analysis, but are still limited in structural complexity and diversity. To improve articulation diversity and realism, ACD [14] collects complex articulated objects from ABO [7], 3D-Future [8] and HSSD [15]. While the articulation structures in ACD are more intricate, the scale of dataset remains limited. To address both diversity and scalability limitations, we present PM-X, a large-scale, URDF-compatible dataset of procedurally generated articulated objects with high structural complexity.

## 3 Generate Articulated Objects from Dual-Image Pairs

### 3.1 Overview

We propose a diffusion network to generate all the parameters of articulated objects conditioned on a pair of dual-state images and a part-level connectivity graph. The overall architecture is illustrated in Figure 2. To support this generation process, we parameterize each part in terms of its spatial location, articulation connectivity, and semantic attributes. The $i$-th part $\mathbf{p}_i$ is represented by the

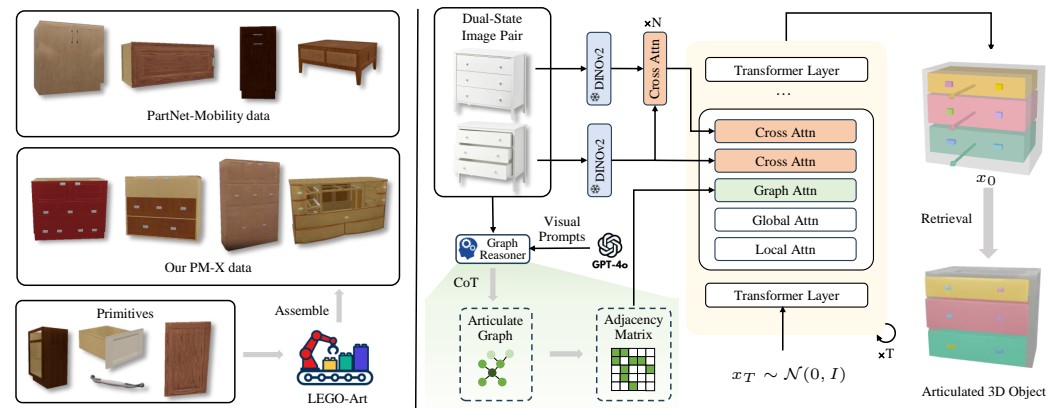

Figure 2: Overview of the proposed DIPO framework. The left part shows the proposed LEGO-Art pipeline assembles the primitives in existing dataset and construct the PM-X dataset, which are more diverse and complex compared to PM dataset. The right part shows that our diffusion model equipped with CoT-based Graph Reasoner for articulate graph inference, and conditioned on resting & articulated image pairs to generate articulated objects.

bounding box coordinates $\mathbf{b}_i \in \mathbb{R}^6$, semantic label $l_i$, articulation type $t_i$, joint axis $\mathbf{a}_i \in \mathbb{R}^6$, and motion range $\mathbf{r}_i \in \mathbb{R}^2$. To facilitate unified processing, all attributes are repeated to a 6-dimensional array, resulting in a $5 \times 6$ matrix representation for each part.

## 3.2 Dual-State Image Conditioning

We condition the denoising process on both resting-state and articulated-state images to capture motion-aware cues. Let $\mathcal{F}_R$ and $\mathcal{F}_A$ denote the DI-NOv2 [29] features from the resting and articulated images, respectively. To integrate these into the diffusion network, we apply a **Dual-State Injection Module** at each layer.

Given part embeddings $X$, we first perform cross-attention with resting-state features $\mathcal{F}_R$ to capture static appearance. We then guide articulated features $\mathcal{F}_A$ to attend to $\mathcal{F}_R$, and subsequently inject this context-enhanced signal into $X$. The overall conditioning update at each diffusion step is defined as:

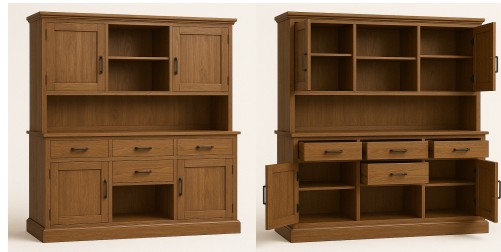

(a) Resting state      (b) Articulated state

Figure 3: Dual-state visual prompt used by the *Graph Reasoner*. GPT-4o can produce realistic and structurally complex image pairs.

$$X = X + \mathrm{CA}(X, \mathcal{F}_R) + \mathrm{CA}(X, \mathrm{CA}(\mathcal{F}_A, \mathcal{F}_R)), \tag{1}$$

where $\mathrm{CA}(Q, K)$ denotes a standard cross-attention operation that query $Q$ attends to key-value source $K$. This design allows the model to generate more accurate part movement and joint behavior by contrasting the two input states.

## 3.3 Graph Reasoner via Chain-of-Thought Prompting

We introduce the **Graph Reasoner**, a Chain-of-Thought (CoT) based module that predicts the articulated part connectivity graph from dual-state images, serving as a structural prior for the diffusion process. The reasoning follows a step-by-step paradigm. It first identifies candidate parts and estimates their coarse spatial layout, then verifies whether the layout satisfies the given articulation rules, and finally infers attachment relationships to generate the articulation graph. After that, we convert the predicted articulation graph into an adjacency matrix, which serves as an attention mask to guide the self-attention of the diffusion model along valid structural connections.

In addition, we leverage the instruction-following and visual-editing capabilities of GPT-4o to generate dual-state image pairs of structurally diverse objects as Figure 3 shows. These results

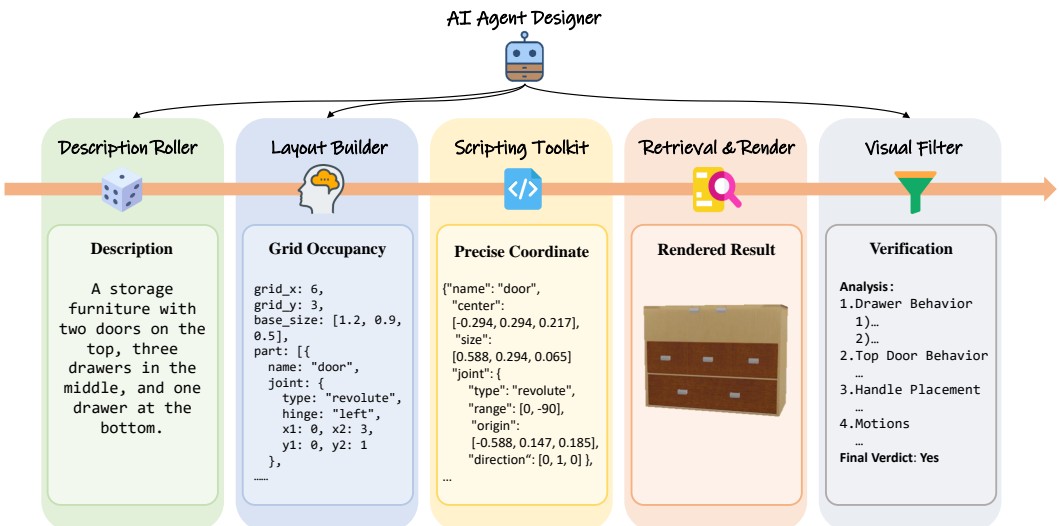

Figure 4: An overview of the fully automated synthesis pipeline for the proposed PM-X dataset. The synthesis pipeline consists of five functional modules executed in sequence: (1) a description roller that uses an LLM to generate natural language descriptions for structured layout, (2) a layout builder to generate part-level grid occupancy and joint configurations, (3) a scripting toolkit to construct precise coordinate from the grid-based layout information, (4) a retrieval and render module to assemble geometry and render dual-state images, and (5) a visual filter that uses a VLM to validate the plausibility of generated samples. In particular, modules (1), (2), and (5) are automatically constructed and managed by the AI Agent Designer.

serve as strong example visual prompts for the Graph Reasoner to achieve a higher stability and generalization of graph prediction.

## 4 Construct Complex Data from Partnet-Mobility

### 4.1 LEGO-Art pipeline

To gain favorable performance on challenging data, we require a large-scale 3D dataset with diverse part layouts. However, existing datasets still fall short in complementary ways: PM [48] offers sufficient data but lacks articulation complexity, while ACD [14] includes more realistic kinematic structures but is limited in dataset scale.

To address this issue, we design a **L**anguage-driven **E**ngine via **G**rid **O**rganization for **Art**iculation objects construction (**LEGO-Art**). It is a fully automated synthesis pipeline that generates complex articulated 3D assets by assembling part primitives from existing dataset. Figure 4 shows the overall workflow of the synthesis pipeline. The details of each step are illustrated below.

- **Description Roller.** The pipeline begins by generating a natural language description of an articulated object by a LLM agent (e.g., "a storage furniture with two doors on top, three drawers in the middle, and one drawer at the bottom"). This serves as a high-level blueprint for the object's structure without requiring precise geometry.

- **Layout Builder.** Given this textual input, the second agent translates the description into a part layout and articulation configuration. Instead of predicting exact 3D coordinates, which often introduces hallucination, we discretize the space into a grid and assign parts to grid cells. Each part is associated with joint metadata such as type, axis, and motion direction.

- **Scripting Toolkit.** We develop a scripting toolkit that converts the grid-level spatial layout into precise 3D coordinates and assigns articulation parameters of the axis and direction of joint, motion range and joint type.

- **Retrieval & Render.** We assign geometry to each part by retrieving mesh primitives from PartNet-Mobility by the algorithm proposed by [24]. Parts are scaled and positioned

according to the layout, and connected as specified by the URDF. Then, we render a resting and articulated state image pair of each object by BLENDER.

- **Visual Filter.** To ensure data quality, we include a final filtering step. We use a VLM to assess whether each rendered object plausibly matches its description and articulates correctly. Only assets that pass this check are included in our final dataset, PM-X.

- **AI Agent Designer.** To simplify the development of the above components, we adopted a prompt-based agent design process. Specifically, we described our intended system behavior in natural language and used an LLM to co-design the system prompts for the Description Roller, Layout Builder, and Visual Filter agents.

The proposed LEGO-Art enables scalable generation of physically valid, semantically rich, and structurally diverse articulated assets with minimal human effort, and plays an essential role in enabling our DIPO to generalize to more challenging dataset.

### 4.2 PM-X Dataset

Based on the LEGO-Art, we build a large-scale dataset from the part primitives of the PartNet-Mobility dataset, named **PM-X**. PM-X consists of 600 automatically generated structural-complex articulated objects. For every object, we futher provide correspondence rendered images, URDF files, and natural language descriptions. Due to the experiments settings, we only consider StorageFurniture and Table objects in the proposed dataset. However, the

Table 1: Comparison of dataset scale and part complexity.

| Dataset | # Objects | Avg. # Parts |
|---------|-----------|--------------|
| PM [48] | 570 | 4.94 |
| ACD [14] | 135 | 7.48 |
| PM-X (Ours) | 600 | 19.40 |

synthesis pipeline can be extended to a wider category of articulated objects, and the overall dataset size can also be scaled up. Compared to existing datasets, PM-X offers not only significantly greater structural complexity and articulation diversity, but also sufficient scale to serve as a standalone training set for generative models. These characteristics make it particularly effective for improving generalization and robustness in articulated object generation tasks, especially under out-of-distribution settings. Our experiments also demonstrate the superiority of the PM-X dataset. Table 1 illustrates that the PM-X dataset surpasses previous datasets in both object quantity and average part count, highlighting its scalability and structural richness.

## 5 Experiments

### 5.1 Implementation Details

We follow the dataset split way of SINGAPO [23] to build the training and testing set. Specifically, the training set is made up of $493$ articulated objects from the PM [48] dataset, combined with $600$ samples from our proposed PM-X dataset. Each object is rendered by BLENDER_EEVEE_NEXT engine to produce dual-state image pairs from $20$ random views. We further introduce a complex data augmentation to enhance the performance of the model, which is detailed in the supplementary materials. For evaluation, we use 77 held-out objects from PM, each rendered from two random views, resulting in 144 dual-state test samples. In addition, we include 135 objects from the ACD dataset [14] to further assess the generalizability of the model to out-of-distribution data.

To accelerate convergence, we initialize our model with the pretrained weights from CAGE [24]. We train our model for 200 epochs with a batch size of 20. The model is optimized by AdamW [26] with $\beta = (0.9, 0.99)$ The learning rate is set to $5 \times 10^{-4}$ for the image-conditioned module and $5 \times 10^{-5}$ for the base model. All experiments are conducted on 8 NVIDIA 4090 GPUs.

### 5.2 Comparisons

#### 5.2.1 Baselines & Metrics

Three representative methods, which are URDFormer [6], NAP [18], and SINGAPO [23], are selected as comparison baselines. Specifically, we finetune the pre-trained URDFormer and retrain the SINGAPO for a fair comparison. For NAP, we follow the experiment setting of SINGAPO that

Table 2: Comparison of reconstruction quality and graph prediction accuracy on **PartNet-Mobility** test set. Lower is better (↓) except for Acc% (↑).

| | Reconstruction quality | | | | | | Graph |
|---|---|---|---|---|---|---|---|
| | RS-$d_{\text{gIoU}}$ ↓ | AS-$d_{\text{gIoU}}$ ↓ | RS-$d_{\text{cDist}}$ ↓ | AS-$d_{\text{cDist}}$ ↓ | RS-$d_{\text{CD}}$ ↓ | AS-$d_{\text{CD}}$ ↓ | Acc% ↑ |
| URDFormer [6] | 1.2327 | 1.2332 | 0.2885 | 0.4403 | 0.4417 | 0.6910 | 6.62 |
| NAP-ICA [18] | 0.5706 | 0.5765 | 0.0563 | 0.2547 | 0.0209 | 0.3473 | 25.06 |
| SINGAPO [23] | 0.5134 | 0.5236 | 0.0487 | 0.1107 | 0.0191 | 0.1270 | 75.97 |
| **DIPO(Ours)** | **0.4561** | **0.4683** | **0.0359** | **0.0732** | **0.0132** | **0.0423** | **85.06** |

Table 3: Comparison of reconstruction quality and graph prediction accuracy on **ACD** test set. Lower is better (↓) except for Acc% (↑).

| | Reconstruction quality | | | | | | Graph |
|---|---|---|---|---|---|---|---|
| | RS-$d_{\text{gIoU}}$ ↓ | AS-$d_{\text{gIoU}}$ ↓ | RS-$d_{\text{cDist}}$ ↓ | AS-$d_{\text{cDist}}$ ↓ | RS-$d_{\text{CD}}$ ↓ | AS-$d_{\text{CD}}$ ↓ | Acc% ↑ |
| URDFormer [6] | 1.1074 | 1.1094 | 0.2868 | 0.3948 | 0.6229 | 0.7608 | 1.52 |
| NAP-ICA [18] | 0.9955 | 1.0000 | 0.1713 | 0.3246 | 0.1141 | 0.3061 | 8.27 |
| SINGAPO [23] | 0.9700 | 0.9728 | 0.1582 | 0.2057 | 0.1047 | 0.1762 | 36.67 |
| **DIPO (Ours)** | **0.9126** | **0.9151** | **0.1253** | **0.1541** | **0.0751** | **0.1085** | **48.15** |

insert an image cross attention block into each layer to achieve controllable generation of images, marked as NAP-ICA.

To evaluate reconstruction quality and articulation correctness, we adopt four metrics: (1) $d_{\text{gIoU}}$ ↓, the generalized IoU between predicted and ground-truth part bounding boxes; (2) $d_{\text{cDist}}$ ↓, the Euclidean distance between part centers; (3) $d_{\text{CD}}$ ↓, the Chamfer Distance [2] between predicted and ground-truth meshes; and (4) Acc ↑, the graph prediction accuracy. All metrics are computed over both resting and articulated states. For clarity, we prefix the metric names with RS- and AS- in the tables to indicate the evaluation state.

### 5.2.2 Quantitative Comparison

We report quantitative results on the PM and ACD datasets in Table 2 and Table 3, respectively. To reduce the impact of stochastic variation, we evaluate all diffusion-based generative methods five times per test sample and report the averaged metric values.

As shown in Table 2, our method DIPO achieves the best performance in terms of reconstruction quality and accuracy of articulate graph on the PartNet-Mobility test set. Importantly, we observe that the performance drop from RS (rest ing state) to AS (articulated state) is significantly smaller for our method than for all others. It indicates that dual-image conditioning provides effective control signals that help the model maintain accurate articulation predictions.

On the ACD test set (Table 3), which contains more diverse and realistic articulated objects, our method continues to outperform all baselines. DIPO shows consistently superior reconstruction accuracy in both states and delivers the best graph prediction accuracy. The evaluation results on ACD dataset demonstrate that our method performs well on out-of-distribution data.

The above results demonstrate that the proposed DIPO achieves superior quantitative performance with both high accuracy and strong generalization across structurally diverse datasets.

### 5.2.3 Qualitative Comparison

Figure 5 provides a qualitative comparison between our method and two strong baselines, NAP-ICA [18] and SINGAPO [23]. Each example includes: (1) the input dual-state image pair (closed and open), (2) the predicted articulation graph, (3) the reconstructed part layout and joints in resting state, and (4) the final articulated geometry. The examples cover a wide spectrum of scenarios, including synthetic data from PM and ACD datasets. In addition, the last three rows are real-world examples: we either collect resting-state images from the Internet or directly capture image pairs of nearby objects in both states. This provides a more realistic evaluation of generalization beyond existing datasets. For Internet-collected examples that only provide resting-state images, we employ GPT-4o to generate the articulated counterparts, showcasing the flexibility of our method.

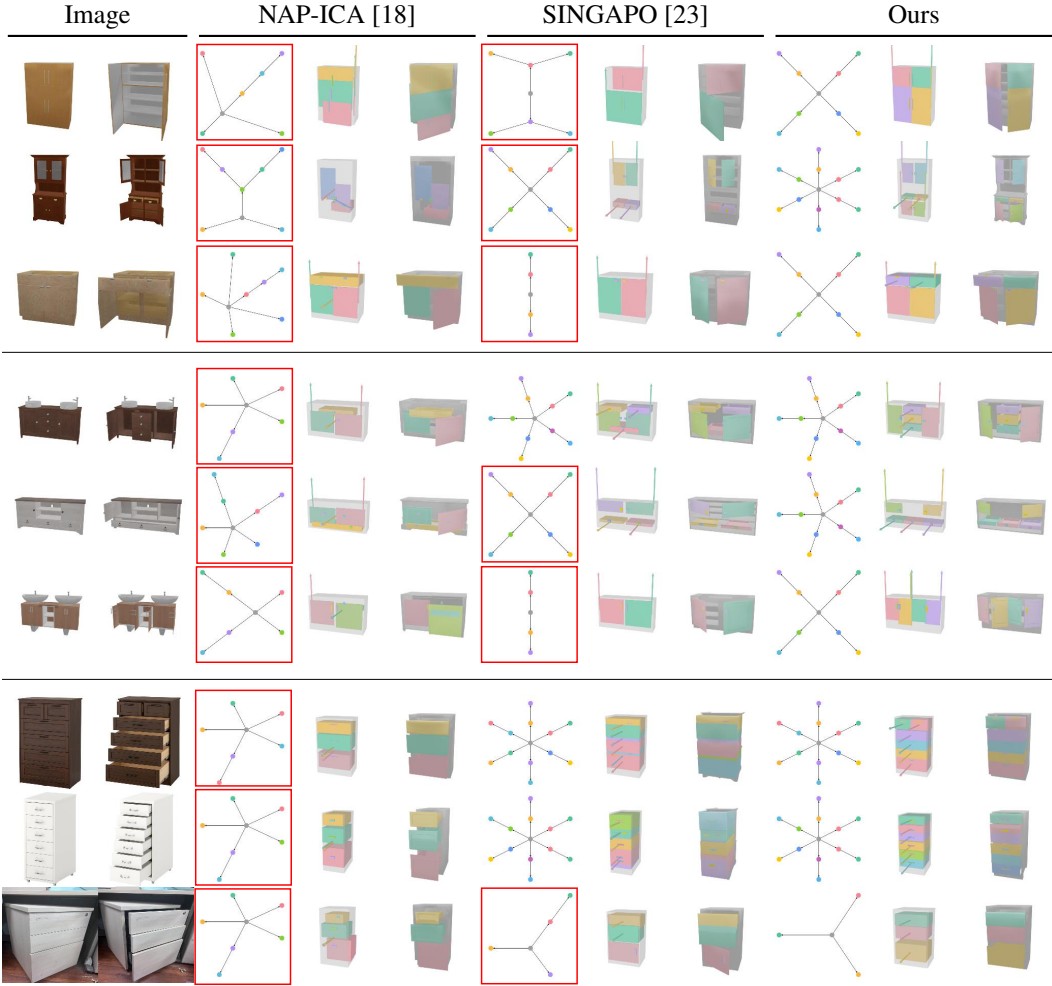

Figure 5: Visual comparison between the proposed DIPO and two baselines. The fist two columns show the dual-state image pairs. The precddiction results of articulate graph, the part layout and joint visualization in resting state, and the final geometry in articulated state are also illustrated. The first three rows are sampled from the PM dataset, the middle three rows are from the ACD dataset, and the last three rows are real-world images. Incorrect parts connections are marked with red box.

Compared to baselines, our method DIPO demonstrates superior visual quality and better accuracy of articulation graph prediction. Thanks to the large-scale structurally diverse training provided by the PM-X dataset, our method shows better robustness when handling complex objects or real-world data. Moreover, cases in which parts are densely arranged and exhibit highly similar textures often confuse single-image baselines, resulting in incorrect articulation inference. In contrast, our method leverages the contrastive cues between resting and articulated states to recognize part boundaries, joint connectivity, and part motions more accurately.

These qualitative results strongly support the effectiveness and generalization ability of the proposed DIPO.

## 5.3 Ablation Study

We conduct detailed ablation studies to verify the effectiveness of each key component in our framework, including the PM-X dataset, Dual-state Injection Module (DIM), and Graph Reasoner (GR). We construct several variants by selectively altering these components. The quantitative results are summarized in Table 5. In addition, we further analyze the settings of each component in isolation in the following paragraphs.

Table 5: Ablative results of reconstruction quality and graph prediction accuracy on **ACD** test set. Lower is better (↓) except for Acc% (↑).

| Settings | | | Reconstruction quality | | | | | |
|---|---|---|---|---|---|---|---|---|
| PM-X | DIM | GR | RS-$d_{\mathrm{gIoU}}$ ↓ | AS-$d_{\mathrm{gIoU}}$ ↓ | RS-$d_{\mathrm{cDist}}$ ↓ | AS-$d_{\mathrm{cDist}}$ ↓ | RS-$d_{\mathrm{CD}}$ ↓ | AS-$d_{\mathrm{CD}}$ ↓ |
| | | | 0.9872 | 0.9900 | 0.1608 | 0.2096 | 0.1083 | 0.1792 |
| ✓ | | | 0.9429 | 0.9464 | 0.1389 | 0.1868 | 0.0849 | 0.1538 |
| | ✓ | | 0.9565 | 0.9589 | 0.1478 | 0.1819 | 0.0924 | 0.1407 |
| | | ✓ | 0.9902 | 0.9931 | 0.1697 | 0.2157 | 0.1208 | 0.1881 |
| ✓ | ✓ | | 0.9212 | 0.9233 | 0.1257 | 0.1589 | 0.0752 | 0.1200 |
| ✓ | | ✓ | 0.9332 | 0.9368 | 0.1391 | 0.1843 | 0.0844 | 0.1439 |
| | ✓ | ✓ | 0.9497 | 0.9515 | 0.1500 | 0.1786 | 0.0973 | 0.1317 |
| ✓ | ✓ | ✓ | 0.9126 | 0.9151 | 0.1253 | 0.1541 | 0.0751 | 0.1085 |

**Impact of PM-X dataset.** Table 5 shows that across various settings of ablative experiments, incorporating the PM-X dataset consistently improves reconstruction quality, indicating its broad effectiveness. To further validate this effect, we additionally experiment with using only 25% and 50% of the PM-X data. As shown in Figure 6, IoU scores for both resting and articulated states degrade steadily as the PM-X ratio increases, confirming the importance of PM-X in enhancing structural accuracy and generalization.

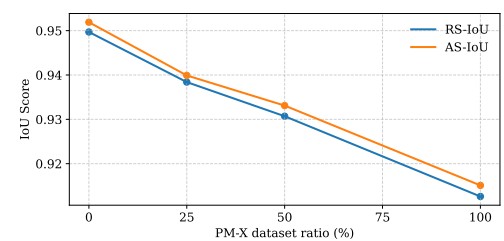

Figure 6: Ablative comparison under different ratios of PM-X data

**Effectiveness of Dual-Image Input.** We conduct ablation experiments to assess the contribution of the DIM module. As shown in Table 5, adding DIM significantly improves performance across all reconstruction metrics. The effectiveness of DIM is further reflected in Figure 1, 5, where our method accurately identifies the motion direction according to articulated images. It demonstrates that the dual-image design not only enhances articulation prediction, but also imposes the ability of structural reasoning to the model.

**Analysis of Graph Reasoner** As illustrated in Table 5, the GR module can not consistently improve performance across all settings. This is because while GR enables more accurate prediction, it also tends to produce more complex topologies. For model variants not trained on the PM-X dataset, such complex graphs may become out-of-distribution, leading to suboptimal perform However, when the model is trained with the structurally diverse PM-X dataset,

Table 4: Ablative results of Graph Reasoner.

| Settings | Acc% ↑ |
|---|---|
| w/o CoT | 39.26 |
| w/o Visual Input | 37.77 |
| w/o dual-state input | 39.63 |
| Full Model (GR) | 48.15 |

the benefits of GR become more apparent. Moreover, we conduct more detail ablative experiments to verify the effectiveness of each component of GR. The results of prediction accuracy can be seen in Table 4.

## 6 Conclusion

We propose DIPO, a framework that advances vision-conditioned articulated object generation under challenging data. We design a diffusion model conditioned on resting and articulated image pairs for articulated 3D object generation, which provides richer part motion information and leads to improved reconstruction accuracy. A Chain-of-Thought graph reasoner is further introduced to enhance part connectivity prediction. In addition, we develop LEGO-Art, an automated pipeline for constructing diverse and complex articulated objects, and contribute PM-X, a large-scale dataset built by the proposed pipeline. Powered by PM-X, our model achieves superior performance and stronger generalization. Extensive experiments validate the effectiveness of each component and the overall advantage of our approach over existing methods.

## Acknowledgments and Disclosure of Funding

Shenzhen Science and Technology Program (JCYJ20240813114237048) "Science and Technology Yongjiang 2035" key technology breakthrough plan project (2024Z120)Chinese government-guided local science and technology development fund projects (scientific and technological achievement transfer and transformation projects) (254Z0102G), Tianjin Natural Science Foundation Project (25ZXRGGX00290, 24JCJQJC00020, 25JCQNJC01390)

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

## Abstract

Our supplementary materials give more details of the proposed DIPO and the experimental settings, which can be summarized as follows:

- The details of LEGO-Art and Graph Reasoner.
- More visual results of the proposed PM-X dataset.
- The details of data augmentation.
- The code and checkpoint of DIPO for inference.
- A video shows some animated visual examples of complex objects.

## A  Details of LEGO-Art Pipeline

In our LEGO-Art pipeline, we design a modular LLM-based [3, 1] framework, where each agent specializes in a distinct subtask. These agents collaborate to generate structured, diverse, and physically plausible articulated object layouts. Below, we detail the system prompt of each agent.

---

**Description Roller:**

You are an expert in generating clear and realistic natural language descriptions of articulated object structures.

Each object must belong to one of the following categories: **['Storage Furniture', 'Table', 'Refrigerator', 'Dishwasher', 'Oven', 'Washer']**.

Your task is to imagine a plausible object structure from one of these categories and describe only its articulated parts in natural language.

The available part types are: **['base', 'door', 'drawer', 'tray', 'handle', 'knob']**. **Note:** "tray" parts are **only** allowed if the object is a microwave.

Each object must contain exactly one implicit **"base"** part, and any number of other parts, depending on the category.

You will be provided with a complexity level:

- **simple:** 1–5 parts, minimal structure.
- **mid:** 6–10 parts, basic spatial layout.
- **complex:** 11 or more parts with more detailed or hierarchical arrangements.

Your output must:

- Only describe the structure of the object: what parts it has, how many of each, and where they are located (e.g., left, right, middle, top, bottom, inside).
- Use precise but simple language in a single sentence.
- **Exclude** any mention of color, texture, material, appearance, or any decorative details.
- Ensure the description is consistent with both the object category and the specified complexity level.

**Example output:** *"A storage furniture with two doors in the middle, one drawer at the bottom, and four drawers on the left and right sides."*
**Important:** Do not include 3D coordinates or structured data. Only output the structural description in plain English.

---

Figure 7: The system prompt of Descrition Roller.

**Layout Builder:**

**Example Question 1:**
The following code is a function that generates a layout from a given object in a grid format.

```python
def sample_base(grid_x, grid_y, base_size):
    # generate the base, and return a coordinate list of grids
    ...

def generate_part_in_grid(base, grid_coords, x1, x2, y1, y2,...):
    # generate coordinates and articulation info of a part
    ...

def generate_layout(info):
    # convert grid-level layout into coordinates and articulation parameters
    base, grid_coords = sample_base(...)
    articulate_tree = [base]

    for part in info['part']:
        part = generate_part_in_grid(...)
        articulate_tree.append(part)

        ...
```

**Example Answer 1:**
You've developed a complete pipeline for procedurally generating articulated object layouts and rendering them visually. The system includes ...

**Example Question 2:**
I need you to generate the info in a python dict from a natural language description. The dict is the only python code in your output. Note that all [x1, x2, y1, y2] should be an integer. The name of the part can only be one of the [drawer, door, handle, knob] (strictly!)

**Example Answer 2:**
Got it! You want to input a natural language description like:
"A wide cabinet, approximately 1.5×1.0×0.5 meters in size, contains two left-hinged doors, each with one handle, and two drawers, each with two handles"
and have it automatically generate a structured info dictionary as:

```python
# python dict
{
    ...
}
```

Figure 8: The system prompt of Layout Builder. This agent is inspired by the code of scripting toolkit and produce a python dict that contains the information of parts layout in grid-level.

---

**Visual Filter:**

You are an expert in **3D object structure verification**.
You will be shown a pair of rendered images of a 3D articulated object: one in the **closed state**, and one in the **open state**. These images are generated based on a predicted structure and joint configuration.

**Your task** is to determine whether the observed articulation behavior is **physically plausible and logically consistent**. That is, check if the object's opening and closing behavior matches how real-world articulated objects work.

You must analyze whether:

- The joints behave correctly (e.g., drawers slide outward, doors rotate from hinges).

- Each handle or knob is correctly positioned and attached to a moving part.

- There are no unreasonable collisions, floating parts, or detached motion.

- The motion (from closed to open) is consistent with the structure and joint types.

**Final Output:** After your analysis, respond with exactly one of the following:

- **Yes** — if the object's motion and structure are physically and functionally plausible.
- **No** — if there are any structural, physical, or semantic inconsistencies.

Figure 9: The system prompt of Visual Filter.

# B Details of Graph Reasoner

The proposed Graph Reasoner can infer articulated connectivity from a dual-state image pair based on chain-of-thought [43, 17] prompt, which is illustrated as followed:

**Graph Reasoner:**

You are an expert in the **recognition**, **structural parsing**, and **physical-feasibility validation** of articulated objects from image inputs.
You will be provided with two rendered images of the same object:

1. A **closed-state image** (all movable parts in their fully closed positions)
2. An **open-state image** (all movable parts in their fully opened positions)

**Your task** is to analyze the object's articulated structure and generate a **connectivity graph** describing the part relationships.

**Workflow:**

1. **Part Detection**
   - Detect candidate parts in the **closed-state image**, optionally using the open-state image to resolve ambiguity or occlusion.
   - Allowed part types: ['base', 'door', 'drawer', 'handle', 'knob', 'tray']
   - Ignore small decorative elements attached directly to the base.
   - There must be exactly one "base"; "tray" is only allowed for microwaves (but not required).

2. **Step-by-Step Reasoning**
   (a) **Part Listing**: List all detected parts and their counts (no attachment inference yet).
   (b) **Validation**: Enforce structural rules:
       - Exactly one base
       - Each door or drawer may have at most two handles or knobs
       - Every handle/knob must be attached to a door or drawer
       - Trays may only appear in microwaves
   (c) **Attachment Inference**: For each non-base part, infer its parent (e.g., "drawer_1 (attached to base)"). Use the open-state image if necessary.
   (d) **Connectivity Graph Construction**: Output a JSON tree where "base" is the root and all other parts are children with proper hierarchy.

   **Example Output:**

   ```
   {
     "base": [
       { "door": [ { "handle": [] } ] },
       { "drawer": [ { "handle": [] } ] }
     ]
   }
   ```

**Final Output:** You **MUST** output a single JSON tree representing the part connectivity of the object. Use the open-state image to enhance accuracy and completeness, but base your interpretation primarily on the closed-state image.

Figure 10: The system prompt of Graph Reasoner.

Moreover, we use GPT-4o [3, 1] to generate dual-state image pairs, which are example visual prompts to make the Graph Reasoner learn how to generate articulated graph in a few-shot manner. The generated image pairs are shown in Figure 11.

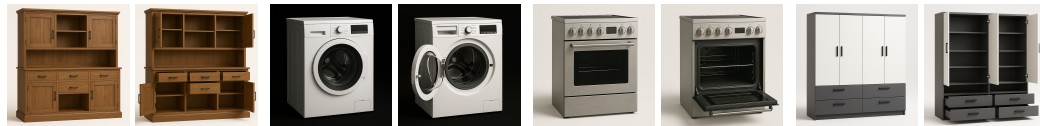

Figure 11: Each pair shows a closed-state image (left) and an open-state image (right) of an articulated object generated by GPT-4o.

## C   More Visual Examples of PM-X

The proposed PM-X dataset provides a large amount of diverse and structurally complex articulated objects. Figure 12 illustrates more visual examples of PM-X dataset. As we can see, each object has a reasonable structure and a rich set of operable parts. In addition, we annotate the description generated by the first stage of LEGO-Art. Objects precisely match the natural language descriptions, enabling LEGO-Art to further serve as a pipeline for text-to-articulated-object generation.

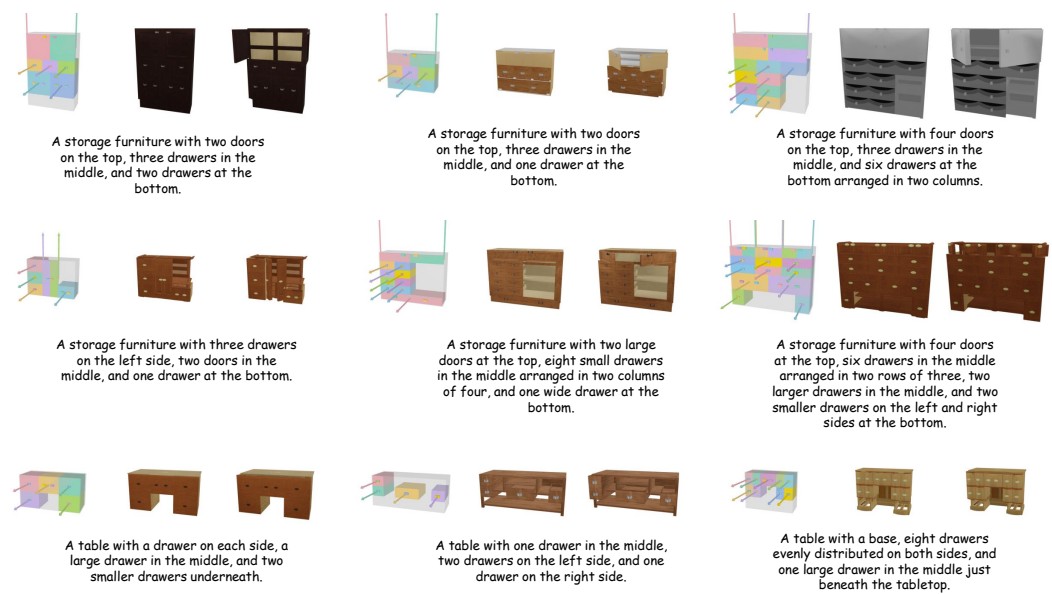

Figure 12: More visual examples of PM-X dataset. Each example includes: (1) the part layout and joints in resting state, (2) a rendered image pair in dual-state, (3) nature language description generated in the fisrt stage of LEGO-Art.

## D   Data Augmentation

We employ several data augmentation during the training state to enhance the robustness and controllability. Data augmentation can be categorized into two types. One focuses on part-level augmentation:

1. Randomly replace small parts like handles and knobs with those of other objects, and perturb their positions.
2. Randomly rescale the whole object.
3. Rotate the whole object upside-down.
4. Stacking several objects together to build more complex objects.

The other one focus on joint-level augmentation:

1. Change the revolute joint into prismatic joint.

2. Randomly modify the direction of revolute joint.
3. Randomly fix the joint.

## E    Limitations & Future Work

We follow the experimental settings of SINGAPO [23] for a fair comparison. However, the benchmark used in SINGAPO only contains several categories, which especially focuses on cabinet-like objects. This limited object diversity may constrain the generalization ability of our model to other articulated categories, such as appliances, tools, or deformable structures. In future work, we plan to build a benchmark that cover a broader range of articulated object types, including both everyday household items and more complex mechanical systems. Our research primarily focuses on predicting more accurate part layouts and joint configurations. We adopt a retrieval-based approach to construct the final 3D objects. Incorporating 3D generation techniques to synthesize more precise and diverse part geometries represents a meaningful direction for future work.

## F    Broader Impact

Our work facilitates controllable generation of articulated 3D objects from dual-state images, enabling structured reasoning over part layout and connectivity. This contributes to downstream applications in embodied AI, virtual environment simulation, and robotics manipulation. By releasing a large-scale synthetic dataset and a modular pipeline, we aim to lower the barrier for research on articulated perception and generation. However, as with any generative framework, care must be taken to avoid misuse such as creating physically implausible or unsafe designs. Moreover, biases in the data distribution or articulation patterns may influence downstream decision-making, highlighting the need for interpretability and robustness in practical deployments.

