# OpenReview forum: "DIPO: Dual-State Images Controlled Articulated Object Generation Powered by Diverse Data"
_NeurIPS.cc/2025/Conference — NeurIPS 2025 poster_

### Official Review · Reviewer_kNAa · 2025-06-30

**Clarity:** 2
**Significance:** 2
**Originality:** 2
**Rating:** 4
**Confidence:** 2

**Summary:**

This paper propose a novel framework, called DIPO, which can controllablly generate 3D object.  Compared to one-image method, this paper utilizes a pair of image to provide more motion information which helps predicting kinematic relationships between parts. This paper also introduce a Chain-of-Thought (CoT)  to causally  infer the relationship of parts. A new dataset, PM-X, is proposed to make a contribution to community.  Moreover,  this paper conduct extensive experiments to demonstrate the outperformance of the proposed method compared to prior arts.

**Questions:**

1. This paper should give more implementation details to help readers easily follow.
2. More objects demos should be given to demonstrate the generalization.

**Ethical Concerns:**

["NO or VERY MINOR ethics concerns only"]

**Final Justification:**

I give 'accept' rate as final decision.

**Limitations:**

This paper should discuss the limitations of the proposed method and the future work.

**Quality:**

2

**Strengths And Weaknesses:**

Strong:
1. This paper propose  a 3D joint object generation framework based on image pairs of the resting state and the extended state. The dual-state image pairs offer important motion information. Compared with the single-image method, the dual-state input solves the fuzziness of component motion and spatial relationship. For multi-view methods, pair images are easy to obtain and cost less computation.
2. To get more accurate relationships of parts, a Chain-of-Thought (CoT) tech is also introduced  to achieve better performance.
3. This paper provides a new dataset, PM-X, which contains diverse and structurally complex joint objects, along with rendered images, URDF annotations, and language descriptions. I thinks the dataset has a great contribution to community.

Weakness.
1. The whole pipeline of this paper is complex, which relays several modules, such as diffusion model, COT and so on. And the implementation  details of this paper is not enough. So, I am afraid it is hard to reproduce the results of this proposed method.
2. This paper just displays some demos about Cabinet.  So, I want to know how about this method perform on other objects. I think it is import to show the Generalization of this paper.

---

> ### Author Rebuttal · Authors · 2025-07-30
>
> We sincerely thank Reviewer kNAa for their constructive and detailed feedback.
> We greatly appreciate the reviewer’s recognition of the technical novelty of our proposed framework, the introduction of the PM-X dataset, and the use of Chain-of-Thought (CoT) reasoning. We also value the reviewer’s suggestions regarding reproducibility and generalization. Below, we provide clarifications and additional evidence to address the concerns raised.
>
> ---
> ### **Q1 Reproducibility Concerns**
>
> We appreciate the reviewer’s concern regarding reproducibility. To facilitate understanding and verification, we provide demo code and pretrained model checkpoints in the supplementary material.
>
> Upon paper acceptance, we will release the full codebase, including all modules (diffusion, CoT-based graph reasoner, layout predictor), as well as the PM-X dataset, trained model weights, and detailed documentation covering implementation, prompt formats, training settings, and data preprocessing steps.
>
> We are committed to ensuring the method is easy to reproduce and build upon.
>
> ---
>
> ### **Q2 Generalization Beyond Cabinet Category**
>
> Although the paper primarily displays *visual* results on cabinet-like objects, our experiments are conducted on **seven object categories** from the PartNet-Mobility dataset: `Storage Furniture`, `Table`, `Refrigerator`, `Dishwasher`, `Oven`, `Washer`, and `Microwave`. All reported quantitative results are evaluated **across these seven categories**. In addition, the ACD dataset provides a finer-grained categorization of object classes. The detailed mapping between PM and ACD categories used in our evaluation is shown below.
>
> **Correspondence Between PM Categories and ACD Categories**
>
> | **PM Category**      | **ACD Category** |
> | -------------------- | ---------------- |
> | StorageFurniture | HangingCabinet   |
> |                      | Cabinet          |
> |                      | Armoire          |
> |                      | ChestOfDrawers   |
> |                      | KitechenCabinet  |
> |                      | Bookcase         |
> |                      | TvStand          |
> |                      | SinkCabinet      |
> | Table            | Table            |
> |                      | Desk             |
> |                      | Nightstand       |
> | Refrigerator         | Refrigerator     |
> | Oven                 | Oven             |
> | Dishwasher           | Dishwasher       |
> | Washer               | Washer           |
> | Microwave            | Microwave        |
>
> We chose to emphasize cabinet-type objects in the qualitative figures because they typically exhibit the most *structural complexity* and better demonstrate the advantages of our approach. However, we agree that including more visualizations from other categories would help illustrate the model’s generalization ability.
> We will add such examples in the revised version to provide a more complete comparison.
>
> Moreover, we report quantitative results across all seven categories in the ACD dataset, as shown in the table below.
>
> **Comparison of DIPO and SINGAPO Across Categories and Metrics (↓ means lower is better)**
>
> | Category         | Method  | RS-\$d\_{gIoU}\$ ↓ | AS-\$d\_{gIoU}\$ ↓ | RS-\$d\_{cDist}\$ ↓ | RS-\$d\_{cDist}\$ ↓ | RS-\$d\_{CD}\$ ↓    | AS-\$d\_{CD}\$ ↓    |
> | ---------------- | ------- | ---------- | ---------- | ---------- | ---------- | ---------- | ---------- |
> | StorageFurniture | SINGAPO | 1.0388     | 1.0421     | 0.1566     | 0.2087     | 0.0983     | 0.1700     |
> |                  | DIPO    | **0.9923** | **0.9946** | **0.1225** | **0.1508** | **0.0684** | **0.0948** |
> | Table            | SINGAPO | 0.9550     | 0.9583     | 0.1670     | **0.1876** | 0.1371     | 0.1628     |
> |                  | DIPO    | **0.8884** | **0.8960** | **0.1254** | 0.1903     | **0.0849** | **0.0971** |
> | Dishwaser        | SINGAPO | 0.5745     | 0.5703     | **0.0911** | **0.1367** | **0.0640** | 0.0958     |
> |                  | DIPO    | **0.4376** | **0.4299** | 0.1229     | 0.1822     | 0.0698     | **0.0948** |
> | Microwave        | SINGAPO | 0.9032     | 0.9027     | **0.1697** | **0.2015** | **0.1141** | 0.1968     |
> |                  | DIPO    | **0.9001** | **0.8999** | 0.1881     | 0.2092     | 0.1164     | **0.1940** |
> | Oven             | SINGAPO | 1.0938     | 1.0948     | 0.1858     | 0.2954     | 0.1949     | 0.2995     |
> |                  | DIPO    | **0.9838** | **0.9861** | **0.1703** | **0.2718** | **0.1584** | **0.2651** |
> | Refrigerator     | SINGAPO | 0.8309     | 0.8304     | **0.1461** | 0.1895     | **0.0881** | 0.1523     |
> |                  | DIPO    | **0.7560** | **0.7556** | 0.1571     | **0.1694** | 0.1161     | **0.1332** |
> | Washer           | SINGAPO | 0.4677     | 0.4732     | 0.1317     | 0.1721     | 0.0775     | 0.1650     |
> |                  | DIPO    | **0.3572** | **0.3645** | **0.0585** | **0.0825** | **0.0248** | **0.0271** |
> ---

---

### Official Review · Reviewer_dwgH · 2025-06-30

**Clarity:** 4
**Significance:** 3
**Originality:** 3
**Rating:** 5
**Confidence:** 4

**Summary:**

This work introduces DIPO, a novel approach to 3D articulated objects generation using dual-image inputs -- one depicting the object in a resting state and the other in an articulated state. Fundamentally, DIPO operates as a diffusion transformer model that infers part layouts and joint parameters from motion cues and spatial relationships derived from the image pair. Initially, the two images are processed through the **Graph Reasoner** module -- a LLM using Chain-of-Thought techniques -- to infer consistent part connection properties in the form of an adjacency matrix. The adjacency matrix, combined with the DINOv2 features extracted from the images, serves as a strong condition signal for the diffusion model using a dedicated cross-attention mechanism. The generated articulated abstraction obtained from the diffusion model is then transformed into an actual articulated asset via retrieval methods (similar to the CAGE approach).

Additionally, to address the scarcity of large-scale datasets featuring high-quality articulated objects, the authors developed an agentic data augmentation pipeline -- named **LEGO-Art** -- and used it to improve the diversity and complexity of PartNet-Mobility dataset -- the current reference dataset for articulated assets. The result is an enriched dataset, termed **PartNet-Mobility-Complex (PM-X)**, where each articulated asset comes with corresponding multi-state rendered images, URDF annotations and a comprehensive language description.

The results are compelling: DIPO outperforms existing methods -- in particular NAP and SINGAPO -- in generating 3D articulated objects and provides enhanced controllability.

**Questions:**

As mentioned earlier, I only have a few minor questions/suggestions:
- Which model did you use as a graph reasoner ? How many billion parameters ?
- As far as I understand, you only tested your approach on the Storage Furniture category.
    - How would you generalize your approach to more complex categories such as, say, robots ?
    - PartNet Mobility contains a bunch of complex, highly redundant assets (e.g. keyboard): how robust is your approach to these assets ? Note that I do not expect new experiment results here: just a few sentences would be enough in my mind to explain this.

- PartNet Mobility contains several assets where one of the mobile parts might be occluded in one of the images (e.g. the tray of a dishwasher): how would DIPO be dealing with this type of assets ? Suggestion: state a few research tracks as future works in your paper.

- What about the assets whose state is not noticeably changing on the two images (e.g. globe with a uniform texture) ?

- In Figure 1, the range of the articulated states seems fully covered, from lower to upper limits. In Figure 5, on the third row, the drawers in the upper and middle assets are only partially opened in the articulated images. Does this affect the joint range of the generated asset ?

- How long does the training take with your setup ? How long does inference takes ?

**Ethical Concerns:**

["NO or VERY MINOR ethics concerns only"]

**Final Justification:**

I appreciate the effort and detailed response provided by the authors in the rebuttal. I believe that my concerns have been well addressed. I therefore maintain my "Accept" rating.

**Limitations:**

yes

**Paper Formatting Concerns:**

- The paper is well written and well presented. The figures are clean, clear and relevant.

- There is a minor typo in Figure 1: DIPO (c) --> DIPO (**b**) and also "if part motion across different articulated states" --> "if part **moves** across different articulated states".

- I believe Figure 2 could be divided in two figures if the size constraints of the paper allow it: typically one figure showing the details of the LEGO augmentation pipeline (maybe just combine it with Figure 4) and one figure showing in greater details the proposed generative pipeline (with maybe further details on the GPT visual prompts and a visual representation of the noisy articulated abstraction at xT, similar to the upper left part of Figure 2 in the CAGE paper).

- In Table 5: please highlight the best two approaches for each column (e.g. best in bold and second best in italic)

**Quality:**

4

**Strengths And Weaknesses:**

### Strengths:
- Strong motivation and related works section: I believe your analysis regarding input modality constraints and limitations in training data -- in the introduction section -- is spot on.
- Additionally, the fact that DIPO uses easily available data (picture of articulated object in two different states) makes it quite handy from a final user perspective. Approaches such as NAP, although quite innovative, are much trickier to control.
- LEGO-Art is definitely a relevant contribution to the field as the lack of large scale curated dataset for articulated objects is quite problematic.
- Strong experiment part
- Relevant ablation study

### Weaknesses:

- The graph reasoner module is insufficiently documented.
- The remaining are only minor considerations detailed in the "Questions" paragraph.

### Conclusion:

This paper is clear, well written and relevant to the field. It is backed by solid experimental evidence and a relevant ablation study. I believe it is up to the standards of NeurIPS: my opinion is that it should be accepted.

---

> ### Author Rebuttal · Authors · 2025-07-30
>
> Thank you for your thorough review and constructive feedback on our manuscript. We appreciate the time and effort you have invested in evaluating our work. Your detailed comments and suggestions have been invaluable in helping us improve the paper. Below, we provide point-by-point responses to address your concerns and questions.
>
> ---
> ### **Q1 Details of Graph Reasoner**
>
> #### **Details of CoT**
> We illustrate the prompt of Graph Reasoner in supplementary materials. We will add more details of step-by-step CoT in revision version.
>
> #### **Add Graph into Diffusion Blocks**
> We convert the predicted articulation graph into a binary adjacency matrix $\mathbf{A} \in \{0,1\}^{N \times N}$, where $N$ is the number of predicted parts. Each entry $\mathbf{A}_{ij} = 1$ indicates the existence of a valid articulated connection from part $i$ to part $j$.
>
> This matrix is used as a structural prior to guide the attention computation in the diffusion model. Specifically, given the latent feature matrix $\mathbf{X} \in \mathbb{R}^{N \times d}$, the attention is computed as:
>
> $$\text{Attn}(\mathbf{X}) = \text{Softmax}\left(\frac{(\mathbf{XW}_Q)(\mathbf{XW}_K)^\top}{\sqrt{d}} + \log(\mathbf{A} + \epsilon)\right) \cdot (\mathbf{XW}_V)$$
>
> where $\mathbf{W}_Q, \mathbf{W}_K, \mathbf{W}_V \in \mathbb{R}^{d \times d}$ are learnable projection matrices, and $\epsilon$ is a small constant (e.g., $1\text{e}^{-6}$) to avoid taking the logarithm of zero. The term $\log(\mathbf{A} + \epsilon)$ serves as a structural attention bias, encouraging the model to focus on physically valid connections during generation.
>
> #### **Details of LLM**
> We use the API of GPT-4o in Graph Reasoner. The parameters of GPT-4o is about 200B acoording to the official report.
>
> ---
>
> ### **Q2 Generalization Ability of DIPO**
>
> #### **Experimental Settings**
> Our experiments include totaling **seven diverse categories** from the PartNet-Mobility dataset, which are `Storage Furniture`, `Table`, `Refrigerator`, `Dishwasher`, `Oven`, `Washer` and `Microwave`. We select these categories to follow the experimental setting of SINGAPO [?] for a fair comparison. There are **no strong category-specific assumptions** on our diffusion model and graph reasoner.
>
> In addition, the ACD dataset provides a finer-grained categorization of object classes. The detailed mapping between PM and ACD categories used in our evaluation is shown below.
>
> | **PM Category** | **ACD Category** |
> |---|---|
> | StorageFurniture | HangingCabinet |
> |  | Cabinet |
> |  | Armoire |
> |  | ChestOfDrawers |
> |  | KitechenCabinet |
> |  | Bookcase |
> |  | TvStand |
> |  | SinkCabinet |
> | Table | Table |
> |  | Desk |
> |  | Nightstand |
> | Refrigerator | Refrigerator |
> | Oven | Oven |
> | Dishwasher | Dishwasher |
> | Washer | Washer |
> | Microwave | Microwave |
>
> #### **Complex Objects**
> However, our method may fail in some extremely complex objects like robots.
> Moreover, assets such as keyboards with a large number of visually similar and spatially redundant parts pose challenges for feature extraction using DINOv2, as the part-level features may lack sufficient discriminability.
> To address this, a promising direction is to incorporate segmentation-aware priors. In particular, we plan to explore integrating SAM-derived segmentation masks to guide part proposal and filtering, enabling more robust articulation reasoning in such densely repetitive structures.
>
> #### **Occlusion and Unchanging Appearance in Dual-State Images**
> In cases where the articulated-state image provides limited additional information due to occlusion (e.g., a dishwasher tray) or minimal appearance change (e.g., a globe with uniform texture), the dual-state input may not fully reveal articulation differences. However, our input setting inherently contains at least as much geometric and structural information as single-image methods like SINGAPO, ensuring a comparable performance lower bound.
>
> To better handle occlusion, we consider incorporating sparse multi-view images (e.g., 2–3 views) as a future extension, which can alleviate visibility issues without incurring significant data collection cost.
>
> In addition, nearly identical dual-state images can still provide useful priors. For example, spherical objects like globes often imply rotation around a central axis, which can guide plausible articulation inference.
>
> ---
>
> ### **Q3 Impact of Partially Opened Articulated Images on Joint Range**
> Our method remains robust to partially opened articulated-state inputs — that is, the predicted joint type and axis remain stable. However, incomplete articulation may influence the estimated joint range, since the model learns this from the observed displacement between states.
>
> For practical use, we recommend using fully opened articulated images to better reveal the maximum joint extent and improve articulation estimation.
>
> To assess the effect of articulation coverage, we additionally render a variant of the test set where the articulated-state poses are randomly sampled (not fully opened), and evaluate our model on this setting. The results are illustrated in the table below.
>
> | **Input Setting** | RS-$d_{\text{gIoU}}$ $\downarrow$ | AS-$d_{\text{gIoU}}$ $\downarrow$ | RS-$d_{\text{cDist}}$ $\downarrow$ | AS-$d_{\text{cDist}}$ $\downarrow$ |
> |---|---|---|---|---|
> | Partially Opened | 0.4987 | 0.5084 | 0.0421 | 0.1031 |
> | Fully Opened | **0.4561** | **0.4683** | **0.0359** | **0.0732** |
>
> ---
>
> ### **Q4 Training and Inference Time**
> Training our full pipeline on eight NVIDIA RTX 4090 GPUs takes approximately **23 hours**.
>
> During inference, we use a pair of $224\times224$ images as input and predict an articulated object with 5 parts. The time consumption of each stage in the inference pipeline is listed in the table below.
>
> | **Stage** | **Time (s)** |
> |---|---|
> | DINO Feature Extraction | 2.63 |
> | Graph Reasoner | 12.62 |
> | Diffusion Model | 3.54 |
> | Retrieval | 8.97 |
>
> ---

---

> ### Comment · Reviewer_dwgH · 2025-08-05
> **Official Comment**
>
> I appreciate the effort and detailed response provided by the authors in the rebuttal. I believe that my concerns have been well addressed. I therefore maintain my "Accept" rating.

---

> > ### Author Response · Authors · 2025-08-08
> >
> > We sincerely appreciate the reviewer’s thoughtful response and are grateful for the positive feedback. We are pleased to hear that our rebuttal has addressed your concerns. Thank you once again for your time and consideration.

---

### Official Review · Reviewer_r4pp · 2025-07-02

**Clarity:** 3
**Significance:** 2
**Originality:** 2
**Rating:** 4
**Confidence:** 4

**Summary:**

This paper proposes DIPO, a framework for generating articulated objects conditioned on a pair of images representing different articulation states. DIPO comprises three main components: (1) a diffusion model that synthesizes articulated objects conditioned on paired images and connectivity graphs; (2) a graph reasoner that predicts connectivity graphs; and (3) an automated agentic data expansion pipeline, LEGO-Art, which leverages VLMs to generate articulated objects as training data. The authors collect a dataset using the proposed data expansion pipeline and conduct experiments to validate the performance of the overall framework.

**Questions:**

1. For the PM-X objects generated by LEGO-Art, is there any manual filtering involved after generation? What is the success rate of the VLM visual filtering?
2. Please also see weakness

**Ethical Concerns:**

["NO or VERY MINOR ethics concerns only"]

**Final Justification:**

My concerns have been addressed by the author. Particularly, (1) there is no mannually filtering, and the success rate is reasonable; (2) the performance of the graph reasoner is pretty good compared to previous methods and (3) the author’s explanation for primarily focusing on boxy object categories is reasonable. Therefore I raised my score to boarderline accept.

**Limitations:**

Yes

**Quality:**

2

**Strengths And Weaknesses:**

Strengths
1. The proposed framework is well aligned with the problem and demonstrates potential for scaling up.
2. The architecture and module design of the diffusion model are well-suited for image-conditioned articulated object generation.
3. The paper presents comprehensive experiments to evaluate the proposed method's performance on articulated object reconstruction and analyze the contributions of individual components.
4. The paper is well written and easy to follow.

Weaknesses
1. Although acknowledged in the limitations section, the experiments are restricted to cabinet-like object categories.
2. The paper does not evaluate the success rate or robustness of the graph reasoner and the data expansion pipeline. Since these components rely heavily on VLMs, the impact of hallucination, biases and ambiguity inherent to VLMs should be assessed.
3. LEGO-Art employs an occupancy grid to represent articulated object structure, which may be effective for boxy, cabinet-like objects but less suitable for those with irregular geometries. Additionally, VLMs are known to have limited spatial reasoning capabilities. Thus, further justification for this design choice is necessary.

---

> ### Author Rebuttal · Authors · 2025-07-30
>
> We sincerely thank Reviewer r4pp for the thorough review and insightful suggestions.
> We appreciate your recognition of the framework’s scalability and the clarity of our methodology and writing. Below we address the concerns regarding evaluation scope, VLM-related robustness, and the design choice behind LEGO-Art in detail.
>
> ---
> ### **Q1 Successful Rate of Filtering**
>
> We do not perform any manual filtering for the PM-X objects generated by LEGO-Art, since they are used exclusively for training data augmentation, where a certain degree of noise is tolerable. While some noisy samples may persist, we empirically find that the large-scale diversity introduced by LEGO-Art still benefits the training of DIPO, and the network learns to ignore occasional outliers.
>
> The LEGO-Art pipeline may fail in two stages:
>
> 1. **Grid Generation Stage**: When converting VLM-generated descriptions into grid-level 3D layouts, hallucinated may lead to invalid or unparseable object layouts. The success rate at this stage is approximately **68.24%**.
> 2. **VLM-based Filtering Stage**: We apply multi-stage visual filtering using CLIP and BLIP to ensure geometric plausibility and articulation consistency. The pass rate of this filtering stage is around **85.11%**.
>
> ---
>
> ### **Q2 Categories Limitation**
>
> Although most visualizations in the main paper focus on cabinet-like categories for clarity, our method is not restricted to cabinet objects. In fact, our experiments include totaling **seven diverse categories** from the PartNet-Mobility dataset, which are `Storage Furniture`, `Table`, `Refrigerator`, `Dishwasher`, `Oven`, `Washer` and `Microwave`.
> In addition, the ACD dataset provides a finer-grained categorization of object classes. The detailed mapping between PM and ACD categories used in our evaluation is shown below.
>
> **Correspondence Between PM Categories and ACD Categories**
>
> | **PM Category**      | **ACD Category** |
> | -------------------- | ---------------- |
> | StorageFurniture | HangingCabinet   |
> |                      | Cabinet          |
> |                      | Armoire          |
> |                      | ChestOfDrawers   |
> |                      | KitechenCabinet  |
> |                      | Bookcase         |
> |                      | TvStand          |
> |                      | SinkCabinet      |
> | Table            | Table            |
> |                      | Desk             |
> |                      | Nightstand       |
> | Refrigerator         | Refrigerator     |
> | Oven                 | Oven             |
> | Dishwasher           | Dishwasher       |
> | Washer               | Washer           |
> | Microwave            | Microwave        |
>
> We select these categories to follow the experimental setting of SINGAPO for a fair comparison. There are **no strong category-specific assumptions** on our diffusion model and graph reasoner.
>
> Moreover, most of the previous work like SINGAPO, URDFormer, CAGE and ACD dataset are focus on openable containers, as these are the prevalent in indoor scenes. However, there are still many issues to be resolved in the construction of articulated openable containers. We believe that the progress in these categories we achieved is meaningful and will be beneficial to the community. We further provide the performance on ACD datset across all categories to demonstrate the generalization ability of DIPO.
>
> **Comparison of DIPO and SINGAPO Across Categories and Metrics (↓ means lower is better)**
>
> | Category         | Method  | RS-\$d\_{gIoU}\$ ↓ | AS-\$d\_{gIoU}\$ ↓ | RS-\$d\_{cDist}\$ ↓ | RS-\$d\_{cDist}\$ ↓ | RS-\$d\_{CD}\$ ↓    | AS-\$d\_{CD}\$ ↓    |
> | ---------------- | ------- | ---------- | ---------- | ---------- | ---------- | ---------- | ---------- |
> | StorageFurniture | SINGAPO | 1.0388     | 1.0421     | 0.1566     | 0.2087     | 0.0983     | 0.1700     |
> |                  | DIPO    | **0.9923** | **0.9946** | **0.1225** | **0.1508** | **0.0684** | **0.0948** |
> | Table            | SINGAPO | 0.9550     | 0.9583     | 0.1670     | **0.1876** | 0.1371     | 0.1628     |
> |                  | DIPO    | **0.8884** | **0.8960** | **0.1254** | 0.1903     | **0.0849** | **0.0971** |
> | Dishwaser        | SINGAPO | 0.5745     | 0.5703     | **0.0911** | **0.1367** | **0.0640** | 0.0958     |
> |                  | DIPO    | **0.4376** | **0.4299** | 0.1229     | 0.1822     | 0.0698     | **0.0948** |
> | Microwave        | SINGAPO | 0.9032     | 0.9027     | **0.1697** | **0.2015** | **0.1141** | 0.1968     |
> |                  | DIPO    | **0.9001** | **0.8999** | 0.1881     | 0.2092     | 0.1164     | **0.1940** |
> | Oven             | SINGAPO | 1.0938     | 1.0948     | 0.1858     | 0.2954     | 0.1949     | 0.2995     |
> |                  | DIPO    | **0.9838** | **0.9861** | **0.1703** | **0.2718** | **0.1584** | **0.2651** |
> | Refrigerator     | SINGAPO | 0.8309     | 0.8304     | **0.1461** | 0.1895     | **0.0881** | 0.1523     |
> |                  | DIPO    | **0.7560** | **0.7556** | 0.1571     | **0.1694** | 0.1161     | **0.1332** |
> | Washer           | SINGAPO | 0.4677     | 0.4732     | 0.1317     | 0.1721     | 0.0775     | 0.1650     |
> |                  | DIPO    | **0.3572** | **0.3645** | **0.0585** | **0.0825** | **0.0248** | **0.0271** |
>
> ---
> ### **Q3 Evaluation of Graph Reasoner**
>
> Some of our statements may have caused you a misunderstanding. The successful rate of Graph Reasoner is illustrated in the Graph Accuracy item in Tables 2 and Table 3. We list them separately in the table below. Moreover, during extensive experiments, we did **not encounter any malformed or structurally invalid graphs** that would cause downstream inference failures.
>
> | **Method**      | **PartNet-Mobility (Acc% ↑)** | **ACD (Acc% ↑)** |
> | --------------- | ----------------------------- | ---------------- |
> | URDFormer       | 6.62                          | 1.52             |
> | NAP-ICA         | 25.06                         | 8.27             |
> | SINGAPO         | 75.97                         | 36.67            |
> | **DIPO (Ours)** | **85.06**                     | **48.15**        |
>
> ---
>
> ### **Q4 Concerns about LEGO-Art**
>
> The LEGO-Art pipeline is proposed to build complex articulated object. In our daily life and experimental settings, we observe that most structurally complex objects that contain many moving parts are in boxy shape, e.g, large storage cabinet. This boxy shape makes them naturally compatible with a grid-based construction scheme, which enables structured, interpretable, and scalable generation.
>
> Motivated by the need to provide supplementary training data featuring such complex structures, we introduce **LEGO-Art** as an automatic pipeline to synthesize articulated objects with diverse part layouts and articulation semantics.
>
> Considering the hallucination issue of VLMs, we do not let the model predict precise continuous coordinates directly, but instead constrain the prediction within discrete grid coordinates.

---

> > ### Comment · Reviewer_r4pp · 2025-08-06
> >
> > Thanks to the authors for the detailed response. My concerns have been addressed. I will raise my score to boarderline accept.

---

> > > ### Author Response · Authors · 2025-08-08
> > >
> > > We are very pleased that our response addressed your concerns and met your expectations. We sincerely appreciate your positive feedback and support for our work.

---

### Official Review · Reviewer_tw3w · 2025-07-05

**Clarity:** 2
**Significance:** 2
**Originality:** 2
**Rating:** 4
**Confidence:** 3

**Summary:**

This paper presents a dual-state image model for controllable articulated 3D object generation, consisting of layout diffusion and CoT-based connectivity reasoning. It also introduces a pipeline to construct structurally diverse articulated objects. Thanks to the introduced pipeline, a new large-scale dataset, PM-X, is proposed with better diversity and complexity.

**Questions:**

###  Questions

1. Could the authors clarify whether the RS- and AS- metrics in Tables 2 and 3 correspond to using only resting-state or articulated-state images? Is there an experiment that evaluates performance when both image types are used together?

2. Have the authors conducted experiments on other articulated object categories beyond StorageFurniture and Table? If not, is there a specific reason why the method is limited to these categories?

3. Is there an evaluation of the model performance on PM-X alone to justify its value?

**Ethical Concerns:**

["NO or VERY MINOR ethics concerns only"]

**Final Justification:**

The reply of the authors addresses most of my concerns. I only have the questions about the PM-X dataset, but this will not limit the contributions of this work. So I raise my score to borderline accept.

**Limitations:**

yes

**Quality:**

3

**Strengths And Weaknesses:**

### Strengths

1. The proposed framework for articulated object generation is technically sound to me. Combining resting-state image and articulated-state image dose provide enriched motion information for predicting kinematic relationships between parts.

2. The proposed automated dataset expansion pipeline and the large-scale dataset of complex articulated 3D objects are helpful for enhancing the performance of articulated 3D object generation.

3. Part of the code is provided.

### Weaknesses

1. Quality. The experimental settings require clarification. In Tables 2 and 3, the meanings of RS- and AS- metrics are ambiguous — do they correspond to models using only resting-state or articulated-state images? It is also unclear how the model performs when both types of images are used together. Additionally, the experiments are limited to StorageFurniture and Table categories, leaving the generalization ability to other articulated objects unverified.

2. Clarity. The writing could be improved for better readability. For example, the AI Agent Designer introduced in Lines 179–182 does not seem to be a direct part of the synthesis pipeline. Also, the placement of Table 4 after Table 5 disrupts the logical flow. Furthermore, the descriptions of the synthesis pipeline and its modules are insufficiently detailed, which hampers understanding.

3. Significance. The contribution of the PM-X dataset is unclear. In experiments, it is only used to enrich training data, without standalone evaluation. Moreover, PM-X includes only two object categories, which limits its broader applicability. A more detailed discussion on its role and value is needed.

4. Originality. The framework for articulated 3D object generation and the synthesis pipeline relies on existing techniques such as LLM, VLM, and retrieval algorithms. This integration, while practical, may reduce the perceived novelty of the approach. Clarifying the unique contributions beyond combining known components would help better establish the work's originality.

---

> ### Author Rebuttal · Authors · 2025-07-30
>
> Thank you for your detailed review and the time you invested in evaluating our work. We appreciate your constructive feedback, which has helped us identify important areas for improvement. We have carefully addressed each of your concerns and believe our responses demonstrate the value and rigor of our approach. Below, we provide point-by-point responses to your questions and comments.
>
> ---
>
> ### **Q1 Clarification on RS-/AS- Metrics**
>
> In all evaluation, our model are consistently conditioned on **both** resting-state and articulated-state images as input.
> The RS- and AS- metrics instead correspond to the **evaluation articulation state**, not the input modality. Specifically:
>
> * **RS-**: We evaluate the predicted 3D object in the **resting state**, and compare it against the ground-truth shape in the same resting state.
> * **AS-**: We evaluate the predicted 3D object in the **articulated state**, and compare it against the ground-truth shape in that articulated configuration.
>
> We will revise the manuscript to make this clearer in both the table captions and main text.
>
> ---
>
> ### **Q2 Clarification on experiment categories**
>
> Our experiments are not limited to only `Storage Furniture` and `Table`. We also include other articulated object categories from the PartNet-Mobility dataset, including `Storage Furniture`, `Table`, `Refrigerator`, `Dishwasher`, `Oven`, `Washer` and `Microwave`, totaling **seven diverse categories**.
> In addition, the ACD dataset provides a finer-grained categorization of object classes. The detailed mapping between PM and ACD categories used in our evaluation is shown below.
>
> **Correspondence Between PM Categories and ACD Categories**
>
> | **PM Category**      | **ACD Category** |
> | -------------------- | ---------------- |
> | StorageFurniture | HangingCabinet   |
> |                      | Cabinet          |
> |                      | Armoire          |
> |                      | ChestOfDrawers   |
> |                      | KitechenCabinet  |
> |                      | Bookcase         |
> |                      | TvStand          |
> |                      | SinkCabinet      |
> | Table            | Table            |
> |                      | Desk             |
> |                      | Nightstand       |
> | Refrigerator         | Refrigerator     |
> | Oven                 | Oven             |
> | Dishwasher           | Dishwasher       |
> | Washer               | Washer           |
> | Microwave            | Microwave        |
>
> This seven categories are selected to follow the experimental setting of SINGAPO \[Liu et al., ICLR 2025], ensuring a fair and direct comparison. We also mentioned this point in supplementary materials.
>
> We further provide the performance on ACD datset across all categories to demonstrate the generalization ability of DIPO.
> We will highlight our experimental settings in the revised version prevent readers from misunderstanding the generalizability of our method.
>
> **Comparison of DIPO and SINGAPO Across Categories and Metrics (↓ means lower is better)**
>
> | Category         | Method  | RS-\$d\_{gIoU}\$ ↓ | AS-\$d\_{gIoU}\$ ↓ | RS-\$d\_{cDist}\$ ↓ | RS-\$d\_{cDist}\$ ↓ | RS-\$d\_{CD}\$ ↓    | AS-\$d\_{CD}\$ ↓    |
> | ---------------- | ------- | ---------- | ---------- | ---------- | ---------- | ---------- | ---------- |
> | StorageFurniture | SINGAPO | 1.0388     | 1.0421     | 0.1566     | 0.2087     | 0.0983     | 0.1700     |
> |                  | DIPO    | **0.9923** | **0.9946** | **0.1225** | **0.1508** | **0.0684** | **0.0948** |
> | Table            | SINGAPO | 0.9550     | 0.9583     | 0.1670     | **0.1876** | 0.1371     | 0.1628     |
> |                  | DIPO    | **0.8884** | **0.8960** | **0.1254** | 0.1903     | **0.0849** | **0.0971** |
> | Dishwaser        | SINGAPO | 0.5745     | 0.5703     | **0.0911** | **0.1367** | **0.0640** | 0.0958     |
> |                  | DIPO    | **0.4376** | **0.4299** | 0.1229     | 0.1822     | 0.0698     | **0.0948** |
> | Microwave        | SINGAPO | 0.9032     | 0.9027     | **0.1697** | **0.2015** | **0.1141** | 0.1968     |
> |                  | DIPO    | **0.9001** | **0.8999** | 0.1881     | 0.2092     | 0.1164     | **0.1940** |
> | Oven             | SINGAPO | 1.0938     | 1.0948     | 0.1858     | 0.2954     | 0.1949     | 0.2995     |
> |                  | DIPO    | **0.9838** | **0.9861** | **0.1703** | **0.2718** | **0.1584** | **0.2651** |
> | Refrigerator     | SINGAPO | 0.8309     | 0.8304     | **0.1461** | 0.1895     | **0.0881** | 0.1523     |
> |                  | DIPO    | **0.7560** | **0.7556** | 0.1571     | **0.1694** | 0.1161     | **0.1332** |
> | Washer           | SINGAPO | 0.4677     | 0.4732     | 0.1317     | 0.1721     | 0.0775     | 0.1650     |
> |                  | DIPO    | **0.3572** | **0.3645** | **0.0585** | **0.0825** | **0.0248** | **0.0271** |
>
> ---
>
> ### **Q3 Clarity of LEGO-Art Pipeline**
>
> We would like to clarify the role of the **AI Agent Designer** (Lines 179–182).
> The AI Agent Designer is not part of the runtime inference pipeline. Instead, it is responsible for crafting high-quality system prompts for the three core agents in the LEGO-Art synthesis pipeline (e.g., the Description Roller, Layout Generator, and Visual Filter). This design phase ensures coherent collaboration among agents, but does not participate in the actual generation process.
>
> For the description of each step, we provide the full system prompts and workflow logic for each agent in the supplementary material. In the revised version, we will incorporate key details into the main paper and expand the descriptions of each module, including their responsibilities and interaction patterns, to enhance clarity and self-containment.
>
> Moreover, we will detail how we design the prompts of each agent with the help of LLMs in a chatting manner in the supplementary materials.
>
> ---
>
> ### **Q4 Contribution of PM-X dataset**
>
> 1. In Sec. I Introduction, we clarify that the motivation of proposing the PM-X dataset is to provide redundant training data of complex objects. Meanwhile, we only use the PM-X for model training.
> 2. Although PM-X currently includes only two categories (StorageFurniture and Table), this was a deliberate choice rather than a limitation of the synthesis method. In seven categories we studied, we found that these two categories exhibit the most complex and diverse articulated structures — such as cabinets with nested arrangements, multi-panel doors, and densely connected subcomponents.
> 3. The role of the PM-X dataset can be demonstrated in Tab.5 and Fig.6. The PM-X improves the performance for each variant in the ACD testing set. Moreover, performance increases when we add more training data from PM-X.
> 4. We further build 100 objects by LEGO-Art pipeline for a standalone evaluation. We compare the performance of DIPO trained with and without PM-X on the validation set. The results show that incorporating PM-X significantly improves the performance on complex objects. The results are illustrated as follow.
>
> **Performance comparison of DIPO trained with and without PM-X on the validation set**
>
> | Method        | RS-\$d\_{gIoU}\$ ↓ | AS-\$d\_{gIoU}\$ ↓ | RS-\$d\_{cDist}\$ ↓ | RS-\$d\_{cDist}\$ ↓ | RS-\$d\_{CD}\$ ↓    | AS-\$d\_{CD}\$ ↓    |
> | ------------- | ------------------ | ------------------ | ------------------- | ------------------- | ------------------- | ------------------- |
> | DIPO w/o PM-X | 1.1789             | 1.1822             | 0.2412              | 0.2959              | 0.1838              | 0.2835              |
> | DIPO w/ PM-X  | 0.9419             | 0.9502             | 0.1687              | 0.2285              | 0.1148              | 0.1749              |
> ---
> ### **Q5 Novelty of LEGO-Art**
>
> While we leverage LLMs and VLMs as foundational tools, our method goes beyond simple integration. A central contribution of our work is a grid-based spatial reasoning framework that overcomes the limitations of direct coordinate prediction from LLMs.
>
> A key innovation of our framework lies in the design of a grid-based layout prediction mechanism. Instead of asking the LLM to generate precise 3D coordinates, which often leads to hallucinations, we constrain the placement of parts to a discrete 3D grid. We then convert the predicted grid layout into accurate object geometry using a custom mapping script. This design enables reliable structure generation while retaining the controllability and flexibility of language-based input.

---

> > ### Comment · Reviewer_tw3w · 2025-08-06
> >
> > Thank you for your detailed response. Your reply has addressed several of my concerns. I have only one question regarding Q4. The choice of including only two categories in the PM-X dataset is based on the observation that these two categories exhibit the most complex and diverse articulated structures. Since the role of the PM-X dataset is to provide redundant training data and improve the generation performance, I hope there are experiments to show that incorporating only these two categories is sufficient for training. I also hope the PM-X dataset can be further expanded in the future to serve as a benchmark for object generation, not just as a supplement to existing datasets for performance improvements. Considering the paper presents other novel ideas and contributions to object generation, the current limitation of the PM-X dataset does not significantly affect the overall value of the paper. I will raise my score to 4.

---

> ### Author Response · Authors · 2025-08-08
>
> We sincerely appreciate the reviewer’s positive feedback and are thrilled that the response has addressed the concerns raised. We are also grateful for the reviewer’s decision to raise the score to 4, acknowledging the contribution of our work.
>
> ---
> Regarding the concerns about the PM-X dataset, we would like to provide additional clarifications and updates:
>
> ### **1. Sufficiency of Two Categories for Training**
> We further train our DIPO with only PM-X and evaluate the performance on the test set of PM-X.
>
> | Training Data        | RS-$d_{\text{gIoU}}$ $\downarrow$ | AS-$d_{\text{gIoU}}$ $\downarrow$ | RS-$d_{\text{cDist}}$ $\downarrow$ | AS-$d_{\text{cDist}}$ $\downarrow$ | RS-$d_{\text{CD}}$ $\downarrow$ | AS-$d_{\text{CD}}$ $\downarrow$ |
> |---------------|-----------------------------------|-----------------------------------|-----------------------------------|-----------------------------------|----------------------------------|----------------------------------|
> | **PM**        | 1.1789                            | 1.1822                            | 0.2412                            | 0.2959                            | 0.1838                           | 0.2385                           |
> | **PM-X** | 0.9864                            | 0.9953                            | 0.1806                            | 0.2449                            | 0.1319                           | 0.2048                           |
> | **PM + PM-X**  | 0.9419                            | 0.9502                            | 0.1687                            | 0.2285                            | 0.1148                           | 0.1749                           |
>
> The results demonstrate that it is enough to achieve a favorable performance with only two categories of PM-X for training. Moreover, the performance will increase when we add the PartNet-Mobility (PM) dataset for training, as simple data is beneficial to model convergence.
>
> However, our LEGO-Art is capable of generating simpler structures, as it does not solely rely on complex data but is flexible enough to handle a wide range of articulated objects, including those with simpler configurations.
>
> ### **2. Future Expansion of PM-X Dataset**
> In line with the reviewer’s suggestion, we acknowledge the potential of expanding the PM-X dataset to serve as an independent benchmark for articulated object generation. We have already added a test set for evaluation.
>
> In future work, we plan to enhance the PM-X dataset by incorporating a broader range of complexity levels, including both simpler and more intricate articulated structures. Additionally, we aim to increase the dataset's size to achieve better generalization and more comprehensive evaluation. To further improve its quality, we will apply detailed manual curation to select high-quality and diverse examples.
>
> These efforts will transform PM-X into a complete and robust benchmark that can be used by the community to evaluate and compare different approaches to articulated object generation. We believe this expansion will significantly contribute to the advancement of the field.
>
> ---
>
> We are excited that our response has been well-received, and we look forward to implementing the suggested improvements. Thank you once again for the thoughtful feedback and your support of our work.

---

### Comment · Area_Chair_w1ee · 2025-08-01
**Discussion kick-off**

Hi everyone,

Thanks for all your hard work on this paper. The submission is currently on 2-2 split ratings.

The discussion period is now open, and I encourage a productive exchange to clarify the remaining open questions. Reviewers: please engage with the authors early in this discussion window. Please don't hesitate to ask for further clarifications. A robust discussion is needed to make an informed decision.

Looking forward to hearing your thoughts. Let's have a productive chat to get this sorted out!

---

### Note · Authors · 2025-08-14

Dear ACs，

We sincerely thank the Area Chair for their support and thoughtful oversight throughout the review process. We are pleased that our rebuttal has successfully addressed the reviewers' primary concerns and that the reviewers expressed **clear support for the acceptance of our paper**. The overall scores have improved significantly **from 3–3–4–5 to 4–4–4–5**, reflecting the positive reception of our clarifications and additional results.

Specifically, we addressed the following key concerns:

1. **Generalization to diverse object categories**:
   Several reviewers raised concerns about whether our model's performance was limited to only *StorageFurniture* and *Table*. In our rebuttal, we clarified that our experiments cover **seven representative articulated object categories**, and we included detailed per-category comparisons. These results demonstrate that our method consistently outperforms the baseline across all categories, validating its generalization ability.

2. **Completeness and role of the PM-X dataset**:
Reviewers questioned the limited scope of the PM-X dataset and its utility. In response, we introduced a dedicated test set for PM-X and evaluated multiple baselines on it, showing its potential as a standalone benchmark. Furthermore, we reported the success rates of each stage in the PM-X synthesis pipeline to provide more details.

We believe these clarifications and additional experiments have significantly strengthened our submission. Our proposed method and the PM-X dataset together provide valuable resources for future research on controllable articulated object generation.

We are grateful for the constructive feedback from all reviewers and look forward to incorporating all improvements in the final version. We once again thank the Area Chair for their guidance and support in facilitating a fair and constructive review process.

---

### Decision · Program_Chairs · 2025-09-17

**Decision:**

Accept (poster)

**Comment:**

This paper introduces DIPO, a framework for generating articulated 3D objects from a pair of images, alongside a new dataset resource, PM-X. The review process for this paper began with a split decision, reflecting valid initial concerns about the work's generalization and the clarity of its contributions. However, an effective author rebuttal and a constructive discussion period successfully addressed these key issues, leading to a consensus for acceptance among the reviewers.

The final recommendation is to accept this paper for a poster presentation. The work presents a technically sound method for a challenging problem and offers a valuable new dataset to the community.

### Summary
The paper makes two primary contributions. The first is DIPO, a framework for controllable 3D articulated object generation. Its key idea is to use a dual-image input, one of the object in a resting state and another in an articulated state, to provide explicit motion cues. The framework uses a diffusion model to generate part layouts and a CoT-based graph reasoner to infer part connectivity. The second contribution is a new dataset of complex articulated objects, PM-X, which is generated via an automated pipeline called LEGO-Art. The authors claim that the DIPO framework significantly outperforms existing baselines and that the PM-X dataset enhances generalization to more complex object structures.

### Strengths
The reviewers converged on several key strengths:
- The use of a dual-state image pair is a simple but clever and effective idea. It provides rich motion information that is difficult to infer from a single image, yet remains a much lower data collection burden than requiring multi-view video.
- The overall architecture, which combines a diffusion model with a CoT-based reasoner, is well-motivated and technically solid for the task of articulated object generation.
- The introduction of the LEGO-Art pipeline and the resulting PM-X dataset is a significant contribution in its own right. It addresses a known limitation in the field—the lack of diverse and complex articulated object data—and serves as a valuable resource for future research.

### Weaknesses
The initial reviews raised several important weaknesses that made this a borderline case:
- The initial experiments were primarily focused on a few "cabinet-like" object categories (StorageFurniture, Table), raising valid concerns about whether the method could generalize to a wider variety of articulated objects.
- The role and value of the new PM-X dataset were not clearly demonstrated. It was used to supplement training data, but its utility as a standalone benchmark was not evaluated, making its significance hard to gauge.
- Some reviewers found the pipeline complex and certain experimental details, such as the performance metrics, to be unclear.

### Reasons for Recommendation
The recommendation for acceptance is based on the paper's solid dual contribution (a novel method and a new dataset) and, most importantly, the authors' highly effective response to the reviewers' initial concerns. The weaknesses, while valid, were shown to be addressable and were largely resolved during the rebuttal period.
The paper is recommended for a poster presentation. The combination of a novel application-focused method and a new dataset makes it a valuable addition to the conference.

### Summary of Discussion and Rebuttal
The review process began with a 2-2 split in ratings. The main points of contention were the method's generalization capabilities and the significance of the new PM-X dataset.
The authors' rebuttal was comprehensive:
- On Generalization: To address the concern about being limited to cabinet-like objects, the authors provided a new, detailed table showing their model's performance across seven diverse categories. These new results demonstrated that DIPO consistently outperforms the baseline across all categories, effectively refuting the initial criticism.
- On the PM-X Dataset: To clarify the dataset's contribution, the authors provided a new set of experiments evaluating their model's performance when trained with and without PM-X. The results clearly demonstrated that including PM-X leads to a significant performance improvement, validating its utility. They also created a standalone test set for PM-X to establish its potential as a future benchmark.
This targeted response successfully resolved the main concerns. Both of the initially more critical reviewers (r4pp and tw3w) acknowledged that their points had been addressed and consequently raised their scores to "Borderline Accept." This transformed a split decision into a consensus that the paper is a valuable contribution.